# Muscle Fitness to Visceral Fat Ratio, Metabolic Syndrome and Ideal Cardiovascular Health Metrics

**DOI:** 10.3390/nu11010024

**Published:** 2018-12-22

**Authors:** Robinson Ramírez-Vélez, María Correa-Rodríguez, Mikel Izquierdo, Jacqueline Schmidt-RioValle, Emilio González-Jiménez

**Affiliations:** 1Centro de Estudios para la Medición de la Actividad Física CEMA, Escuela de Medicina y Ciencias de la Salud, Universidad del Rosario, Bogotá 111221, Colombia; robin640@hotmail.com; 2GICAEDS Group, Faculty of Physical Culture, Sport and Recreation, Universidad Santo Tomás, Bogotá 110311, Colombia; 3Departamento de Enfermería, Facultad de Ciencias de la Salud, Avda. De la Ilustración, 60, University of Granada, Granada 18016, Spain; jschmidt@ugr.es (J.S.-R.); emigoji@ugr.es (E.G.-J.); 4Department of Health Sciences, Public University of Navarre, Navarrabiomed, IdiSNA, CIBER de Fragilidad y Envejecimiento Saludable (CB16/10/00315), Navarre 31006, Spain; mikel.izquierdo@gmail.com

**Keywords:** metabolic syndrome, muscle strength, visceral fat, ideal cardiovascular health, young adults

## Abstract

This study aimed to investigate the association between the muscle fitness to visceral fat level (MVF) ratio and the prevalence of metabolic syndrome (MetS) and ideal cardiovascular health (CVH) metrics among college students. A total of 1467 young adults recruited from the FUPRECOL study (Asociación de la Fuerza Prensil con Manifestaciones Tempranas de Riesgo Cardiovascular en Jóvenes y Adultos Colombianos), were categorized into four quartiles based on their MVF ratio. Muscular fitness was assessed using a digital handgrip dynamometer and visceral fat level was determined through bioelectrical impedance analysis. Ideal CVH was assessed, including lifestyle characteristics, anthropometry, blood pressure, and biochemical parameters. The body weight, waist circumference, body mass index (BMI), fat mass, fat mass index, and visceral fat level were significantly higher in subjects in Q1 (lower MVF ratio) than those in Q2, Q3, or Q4 (*p* < 0.001). The muscle fitness (handgrip and normalized grip strength (NGS)) of the subjects in Q4 was significantly greater than that of those in Q1 to Q2 (*p* < 0.001). Subjects with a medium-high MVF ratio (i.e., 3–4th quartiles) had an odds ratio of 2.103 of ideal CVH metrics after adjusting for age, gender, university, and alcohol intake (95% confidence interval (CI) 1.832 to 2.414; *p* < 0.001). A lower MVF ratio is associated with worse CVH metrics and a higher prevalence of MetS in early adulthood, supporting the hypothesis that the MVF ratio could be used as a complementary screening tool that could help clinicians identify young adults with unfavorable levels of CVH and metabolic risk.

## 1. Introduction

Abdominal adiposity, dyslipidemia, elevated blood pressure (BP), and impaired fasting glycemia have been identified as cardiometabolic risk factors associated with cardiovascular disease (CVD), with the fastest rise being seen in Latin-American countries [1]. In Colombia, Westernized dietary habits and chronic physical inactivity have been proposed as major factors involved in national overweight and obesity trends [2]. For example, in 2010, 51.2% of women and men aged 18–64 years were either overweight or obese [3]. Excess weight was more prevalent in women than men (55.2% versus 45.6%), particularly concerning the obese subgroup (20.1% versus 11.5%).

Research assessing CVD risk in college students aged 18–24 has revealed an alarmingly high prevalence of abnormal risk factor profiles [4]. As many as 33% of young adults are overweight [5], and this excess weight leads to dyslipidemia and increased incidence of metabolic syndrome (MetS) [6] as well as the risk of developing CVD [7]. In addition, excess epicardial adipose tissue has been proposed as a reliable marker of cardiovascular risk since it is associated with insulin resistance, MetS or fatty liver disease [8]. Taking into account that 80% of CVD events are preventable through diet and lifestyle, early detection and intervention are of special interest [9]. Due to the increasing burden of CVD, the American Heart Association (AHA) established the 2020 Strategic Impact Goals to define the concept of ideal cardiovascular health (CVH) as well as the metrics needed to monitor this across populations [10]. The 7 metrics for assessing CVH in adults (age ≥ 20) comprise 4 health behavior markers (current smoking, body mass index, physical activity, and healthy diet score) and 3 physiological health factors (total cholesterol, blood pressure, and fasting plasma glucose levels). 

Very few studies have analyzed this ideal CVH in young adults, but these suggest a direct association with markers related to future cardiovascular events such as carotid intima-media thickness [11] or left ventricular structure and function later in life [12]. Similarly, Hruskova et al. demonstrated for the first time that CVH is associated with epicardial adipose tissue [13]. In line with this, muscle fitness has been identified as a predictor of cardiometabolic diseases and mortality [14,15]. The physiological mechanisms underlying the relationship between muscle fitness and cardiometabolic risk are fully understood, but muscle mass has been proposed as the main driver of their relationship, as skeletal muscle is the principal site of insulin-mediated glucose uptake, and this is closely related with muscle fitness [16]. Moreover, although low muscle fitness has been associated with CVD risk, studies assessing the independent role of the other components of fitness considered these to be confounders and therefore factors to be controlled in multivariate models [17,18].

Poor diet, obesity, and low physical fitness levels have been shown to be the leading CVD risk factors in Colombian adults, raising concerns about whether an increase in the risk level of these conditions also affects cardiometabolic status [19,20]. These changes have resulted in an increase in the prevalence of overweight and obesity in Colombians, particularly young adults [21]. Additionally, the prevalence of ideal CVH has not been examined in any Latin-American college student population. It is very important to identify the risk factors and take steps to control non-communicable diseases in Colombia. 

Low muscle fitness and high levels of visceral fat tissue have separately been shown to increase the risk of metabolic disorders and CVD early in life [17,22]. However, the potential influence of muscle fitness and visceral fat level together has not been investigated. Therefore, the aim of the present study was to evaluate the association of reduced grip strength (normalized grip strength) as a proxy measure for muscle fitness and physical health and increased visceral fat level mass ratio and the prevalence of MetS and ideal CVH metrics among college students. 

## 2. Methods

### 2.1. Participants and Study Design

This was a secondary analysis of cross-sectional of data from the FUPRECOL study (Asociación de la Fuerza Prensil con Manifestaciones Tempranas de Riesgo Cardiovascular en Jóvenes y Adultos Colombianos), collected between 2014–2017, which aimed to assess changes in lifestyle and CVD during attendance at university (aged 18–30 years) [20]. Details of the study design, characteristics of participants and clinical assessments have been published elsewhere [23,24,25]. A total of 1838 collegiate students from three distinct areas of Colombia—the capital district of Bogota, Boyacá, and Santiago de Cali—were enrolled to participate in the study. Before the recruitment process, collegiate students attended an information meeting, where they were informed of the purpose and procedures of the study. The gender distribution was similar to that of the entire university population. The final sample included in this paper was 1467 (64.2%, 942 women) participants with full valid data. All participants provided written consent, and each study was approved by the local authorized institutional review boards (Bogotá UMB Code N° 01-1802-2013, UR Code N° CEI-ABN026-000010; Cali UNIAJC Code N° 111-02.01.48/16; Tunja Code N° RECT 60) and complied with the Declaration of Helsinki (World Medical Association for Human Subjects). The students who agreed to participate and who had signed the informed consent form were given appointments for the following procedures.

### 2.2. Anthropometric and Body Composition

After completing another general information questionnaire, participants were instructed to wear light clothing (for example, a t-shirt and shorts) for the physical exam. Once the subjects were barefoot and in their underwear, their body weight (kg) was measured using an electric scale (Model Tanita^®^ BC-420-MA^®^ Tokyo, Japan) with a range of 0 to 200 kg and with an accuracy of within 100 g. Height was measured with a portable stadiometer with a precision of 0.1 mm and a range of 0–2.50 m (Seca^®^ 213, Hamburg, Germany). Body mass index (BMI) was calculated by using the formula proposed by Quetelet, where BMI = body mass (kg)/height (m^2^). Body mass index status was evaluated according to the World Health Organization criteria (World Health Organization, 2000) criteria (normal: 18.5 to 24.9 kg/m^2^; overweight: 25.0 to 29.9 kg/m^2^; and obese: ≥30 kg/m^2^) [26]. Waist circumference (WC) was determined by the average of two measurements taken with tape (Lufkin W606PM^®^, Parsippany, NJ, USA) at the waist (at the midpoint between the last rib and the iliac crest). The morphological evaluation process was carried out by a team of professionals (4 physical therapy professors) with extensive experience in anthropometric measurement. Anthropometric variables were measured in accordance with the International Society for the Advancement of Kinanthropometry (ISAK) guidelines [27]. Two percent of the sample was measured twice in order to ensure quality of measures. The technical error of measurement values were less than 2% for all anthropometric variables.

Body fat percentage (BF%), visceral fat level, skeletal muscle mass (kg) and fat mass (kg), were determined for bioelectrical impedance analysis (BIA) by a tetrapolar whole body impedance (Model Tanita^®^ BC-420-MA^®^ Tokyo, Japan). This single-frequency foot-to-foot BIA device provides estimated values for BF% by subtracting the weight of fat-free mass from the total body water. The impedance between the two feet was measured while an alternating current (50 kHz and ~200 μA) passed through the lower body [28]. BIA has been reported to be a valid and reliable method of estimating body composition [29]. For the calculation of intra–inter-observer technical error of measurement, at least 50 subjects needed to be measured, and 48 adults participated (54% women). The corresponding intra-observer technical error (% reliability) of the measurements was 95%. A detailed description of the BIA technique can be found elsewhere [30]. Then, the fat mass index (FMI) was calculated by dividing each subject’s fat mass (kg) by the square of his/her height (m), as previously described [31].

Grip strength as a “proxy” measure for muscle fitness and physical health was assessed using an adjustable digital handgrip dynamometer TKK-18 digital Grip-D dynamometer (Takey^®^, Tokyo, Japan) (range: 5–100 kg; precision: 0.1 kg). It was performed to measure the participant’s maximum force of handgrip. The grip span of the dynamometer was adjusted to the hand size of the participant. With the elbow in full extension, the participant had to press the dynamometer with the right hand for at least 2 s. The test was then repeated with the left hand, performed twice, and the maximum score for each hand was recorded in kilograms. The average of the maximum scores for both hands was used in analyses. As there is substantial covariance between strength capacity and body mass, and the link between strength and both physical function and chronic health is directly mediated by the proportion of strength relative to body mass, handgrip strength was normalized as grip strength (NGS) per body mass; i.e., (handgrip strength in kg)/(body mass in kg). The reproducibility of our data was R = 0.96. Intra-rater reliability was assessed by determining the intraclass correlation coefficient (0.98, CI 95% 0.97–0.99, *n* = 20, median age = 22.8 ± 1.4 years, 66.2 ± 5.4 kg, 1.67 ± 0.1 m, 24.9 ± 3.1 kg/m^2^).

### 2.3. Ideal CVH Behaviors

Data on smoking were collected via self-reported questionnaires (number of cigarettes smoked per day). Ideal smoking status was determined as non-smoker or quit smoking ≥12 months. A standardized questionnaire, the “FANTASTIC” lifestyle (family, physical activity (PA), nutrition, tobacco toxins, alcohol, sleep/stress, personality type, insight, career) questionnaire, was used to collect comprehensive information about substance use via a personal interview with participants [32]. The physically active category was defined as ≥150 min per week of physical activity. Alcohol consumption was defined as subjects who had consumed any alcoholic beverage ≥1 times/week.

A seven-day recall was the dietary assessment tool used to complete the MetDiet adherence survey. The total score was divided into two categories of Mediterranean diet quality: (1) ≤7 points = poor diet quality; and ≥8 points = good diet quality (optimal Mediterranean diet style). Participants who had at least ≥8 points were categorized as having an ideal healthy diet, whereas collegiate students with 7 points were classified as having a non-ideal healthy diet.

### 2.4. Ideal CVH Risk Factors

For blood measurements, the participants were asked to arrive in a fasting state, abstain from exercise training, caffeine, nicotine, and alcohol 12 h before the clinical examination, and continue their regular medication routines. Capillary blood samples (40 µL) were collected to determine serum biochemical parameters, including fasting glucose and total cholesterol using portable Cardiocheck^®^ equipment (Mexglobal SA, Parsippany, NJ, USA).

Blood pressure was taken with an automatic monitor (Omrom® HEM 705 CP/Omron M6 Comfort (Omron® Healthcare Europe B.V., Hoofddorp, the Netherlands) following the recommendations of the European Heart Society (on the right arm, with participants in a supine position and after 10 min of rest) (ESH/ESC Task Force for the Management of Arterial Hypertension, 2013). During the measurement, the participants were seated with their arm supported at the level of the heart. 

### 2.5. AHA Criteria

The AHA guidelines [10] were used to construct an ideal CVH index based on the 7-metrics using the cut-off points for adults, with the participants receiving one point for the presence of each ideal metric. The ideal behaviors defined by the AHA were as follows: BMI < 25 kg/m^2^, physical active individuals (participants who exercised ≥150 min of moderate activity per week), non-smoking status (either never having smoked or having quit smoking >12 months ago), and consumption of a dietary pattern that promotes ideal CVH. Some modifications of the AHA recommendations were required to evaluate diet. Participants who had at least ≥8 points in the MetDiet questionnaire were categorized as having an ideal healthy diet, whereas collegiate students with 7 points were classified as having a non-ideal healthy diet. The clinical and laboratory parameters were classified as an untreated systolic blood pressure <120 mmHg and diastolic blood pressure <80 mmHg, untreated total cholesterol ≤200 mg/dL, and untreated fasting blood glucose <100 mg/dL.

Finally, the participants were categorized into 1 of 3 health levels based on the number of CVH metrics in the ideal range that they exhibited: the healthiest level (favorable ideal CVH score) was defined as having between 5 and 7 metrics in the ideal range; the intermediate level, 3 to 4 metrics; and the unfavorable level, 0 to 2 metrics. These cut-off points have been used in prior international studies [33].

### 2.6. Diagnoses of Metabolic Syndrome

Participants were considered to have a diagnosis of MetS if they had three or more of the following: (1) abdominal obesity (WC ≥ 80 cm in females and ≥90 cm in males); (2) hypertriglyceridemia (≥150 g/dL); (3) low high density lipoprotein cholesterol, HDL-c (<50 mg/dL in females and <40 mg/dL in males); (4) high blood pressure (systolic blood pressure ≥130 mmHg or diastolic blood pressure ≥85 mmHg); (5) high fasting glucose (≥100 mg/dL). MetS was defined in accordance with the updated harmonized criteria of the IDF [34].

### 2.7. Statistical Analysis

Descriptive characteristics are provided as means, 95% confidence Interval (CI), and percentages. Both statistical (Kolmogorov–Smirnov test) and graphical methods (normal probability plots) were used to examine the fit to a normal distribution for each continuous variable. The categorical variables were compared using the Chi squared test. To assess the relationship between muscular fitness and visceral fat ratio and the number of CVH metric, MetS and cardiometabolic risk factors, all study subjects were divided according to quartiles of the MVF ratio (first quartile (Q1 lowest group), second quartile (Q2), third quartile (Q3) and fourth quartile (Q4 highest group). The association between MVF and ideal CVH metrics, as well as with ideal CVH behaviors and factors separately, was assessed by ANCOVA. Finally, logistic regression (odds ratio (OR)) models were employed to compare the prevalence of medium-high MVF (i.e., 3–4th quartiles) across a number of ideal CVH metrics (poor (0–2 metrics), intermediate (3–4 metrics) and ideal (5–7 metrics)) after adjusting by age, gender, university and alcohol use. CVH metrics were also examined as a continuous variable, considering the OR per a 1-metric-higher overall profile. Thus, association between MVF across ideal CVH behaviors (smoking, body mass index, physical activity, and Mediterranean diet adherence), ideal CVH factors (total cholesterol, blood pressure, and plasma glucose) and 7-metric cardiovascular health in college students were analyzed after adjusted by age, gender, university and alcohol use. Statistical analyses were performed using SPSS-IBM (Software, v.24.0 SPSS Inc., Chicago, IL, USA). A *p* value of <0.05 was defined as statistically significant.

## 3. Results

As shown in Table 1, comparisons between Q1, Q2, Q3, and Q4 show that when compared to the other three quartiles, Q4 had a higher prevalence of physical activity, body mass index (BMI) <25 kg/m^2^, a healthy diet, and fasting glucose <100 mg/dL. The proportions of subjects with MetS were 32.3% in Q1, 5.0% in Q2, 1.9% in Q3, and 2.2% in Q4, which were strongly negatively correlated with the MFV ratios in the respective quartiles.

There was an increasing trend of body weight, WC, and BMI from Q1 to Q4 (*p* for trend <0.001). Skeletal muscle mass was significantly lower in Q1 than in Q3, while fat mass, fat mass index, body fat, and visceral fat level were significantly higher in Q1 than in Q2, Q3, or Q4 (*p* for trend <0.001). The muscle fitness (handgrip and NGS) of the subjects in Q4 was significantly greater than that of the subjects in Q1 to Q2 (*p* for trend <0.001).

Figure 1 shows the prevalence of each ideal CVH metric according to the MVF ratio quartiles. The category containing 5–7 metrics was 11.1% in Q1, 52.0% in Q2, 64.2% in Q3, and 65.1% in Q4. In addition, the proportions of subjects with 3–4 metrics were 70.5% in Q1, 45.8% in Q2, 34.8% in Q3, and 34.6% in Q4. Finally, the category containing 0–2 metrics was 18.4% in Q1, 2.2% in Q2, 1.0% in Q3, and 0.3% in Q4. Each of these indicators of improved cardiovascular health was worse where there was a lower MVF ratio (Figure 2). The logistic regression analysis showed that subjects with a medium-high MVF ratio (i.e., 3–4th quartiles) had an OR of 2.103 of ideal CVH metrics after adjusting for age, gender, university, and alcohol intake (95% CI 1.832 to 2.414; *p* < 0.001).

## 4. Discussion

The aim of this study was to investigate the relationship between MVF ratio and the prevalence of MetS and CVH metrics in a large population of 1467 Colombian young adults. The findings indicate that those subjects positioned in Q4 present better CVH metrics, they show a greater proportion of normal BMI values, are more active, and have a healthier diet, as well as having total cholesterol and glucose levels within the normal range. Consistent with these results, the participants in Q3 to Q4 showed the lowest prevalence of metabolic syndrome (1.9 to 2.2%). In contrast, those subjects who were stratified into Q1 presented, in general, a higher number of altered CVH metrics, together with a higher prevalence of MetS (32.3%). Several studies have shown that subjects who exhibit greater numbers of these healthy lifestyle components (HLCs) are more likely to reach later adulthood with lower blood pressure, cholesterol, and blood glucose levels, and favorable intermediate CVD markers [35,36,37].

Muscle fitness and visceral adipose tissue have been independently associated with cardiometabolic health; however, to the best of our knowledge, the association between MVF ratio and CVH has not been examined previously. Our results show that each of the indicators of improved cardiovascular health are worse with a decreased MVF ratio. Overall, these findings support the hypothesis that the MVF ratio could be used as a complementary assessment measure that could help clinicians identify young adults with a low CVH score. In addition, adopting a lifestyle that does not result in a decreased MVF ratio is important in order to prevent unfavorable CVH levels and a higher prevalence of MetS in early adulthood. In this sense, our findings indicate that the MVF ratio is a relevant and effective indicator for defining CVH metrics in Colombian young adults. Therefore, these findings add to the literature supporting the importance of healthy lifestyles for maintaining CVH [38], and reducing atherosclerosis [37], CVD events [39], and mortality [40,41].

On the other hand, the relationship between muscle fitness and the prevention of chronic disease in adult populations is widely recognized [42,43,44]. Evidence suggests that muscle mass and muscular fitness decrease progressively after the age of 20 [45]. Therefore, early adulthood seems to be a crucial timeframe for monitoring and intervention. Muscle fitness has been recognized as a predicting factor for cardiometabolic diseases [46]. Indeed, Steene-Johannessen et al. [47], in a study of 2818 adolescents from Norway, concluded that there is an inverse association between muscle fitness and cardiometabolic profile and MetS risk.

In addition, growing scientific evidence suggests that visceral adipose tissue linked to insulin resistance may play a relevant role in the development of cardiometabolic risk and MetS [48,49,50,51,52,53,54]. In line with this, Cho et al. [49], concluded that the highest quartiles with regard to visceral adipose tissue exhibit a higher risk of MetS as compared with the lower quartiles. Therefore, the visceral fat ratio is an essential tool for estimating cardiometabolic risk and identifying young adults with low CVH.

To the best of our knowledge, our study is the first to apply the CVH metrics to a population of young adults from Colombia, a country with significant health inequality and which is in a situation of nutritional transition. In this context, applying the 7 AHA cardiovascular health metrics allows us to find out how these measures are useful for assessing risks at the populational level in Colombia. In clinical practice, the results from the AHA measures could be used to help assess baseline cardiovascular health in Colombian young people and identify areas where interventions should focus, especially with regard to policies and environmental changes.

Certain limitations of this study also should be addressed. First, the cross-sectional design does not allow us to explain causality. Thus, future prospective analysis is necessary to determine any relationship between the MVF ratio and CVH. In addition, the gold standard for the assessment of visceral fat is magnetic resonance imaging (MRI) and computed tomography (CT). However, in this study, visceral fat measurement was assessed by BIA, which, although it is a highly reliable device, provided an indirect measure. Finally, a potential limitation of our study is that we were not able to adjust for unmeasured variables such as eating time interval and frequency that could affect cardiovascular health, muscle fitness and visceral fat deposition [55]. Despite these limitations, the major strength of the study is that the data were collected from a community-based cohort of young adults in Colombia. Furthermore, highly standardized procedures were developed within the FUPRECOL study to avoid measurement bias.

## 5. Conclusions

A lower MVF ratio is associated with worse CVH metrics and a higher prevalence of MetS in early adulthood. Therefore, MVF ratio is an important and efficient indicator for defining ideal CVH in young adults. Consequently, we propose the use of the MVF ratio as a complementary assessment measure that may be useful for clinicians. Taking into account the fact that CVD and MetS in young adults are considered important public health problems, this study is especially relevant since it provides a novel tool for the early identification of these health issues.

## Figures and Tables

**Figure 1 nutrients-11-00024-f001:**
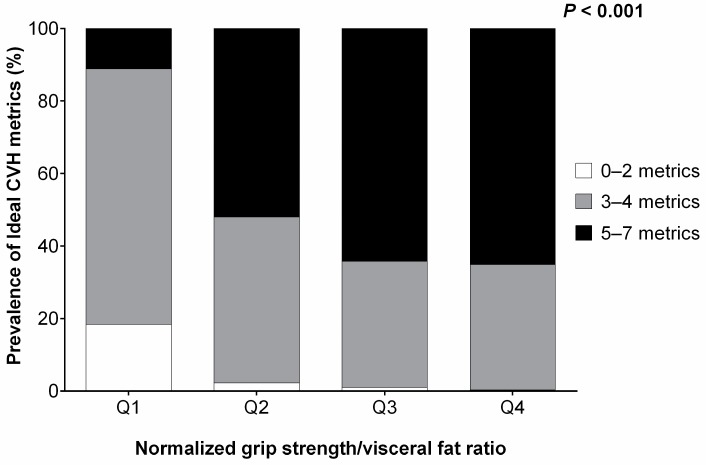
The comparisons of the number of CVH metrics in study subjects with quartile stratification according to the MVF ratio.

**Figure 2 nutrients-11-00024-f002:**
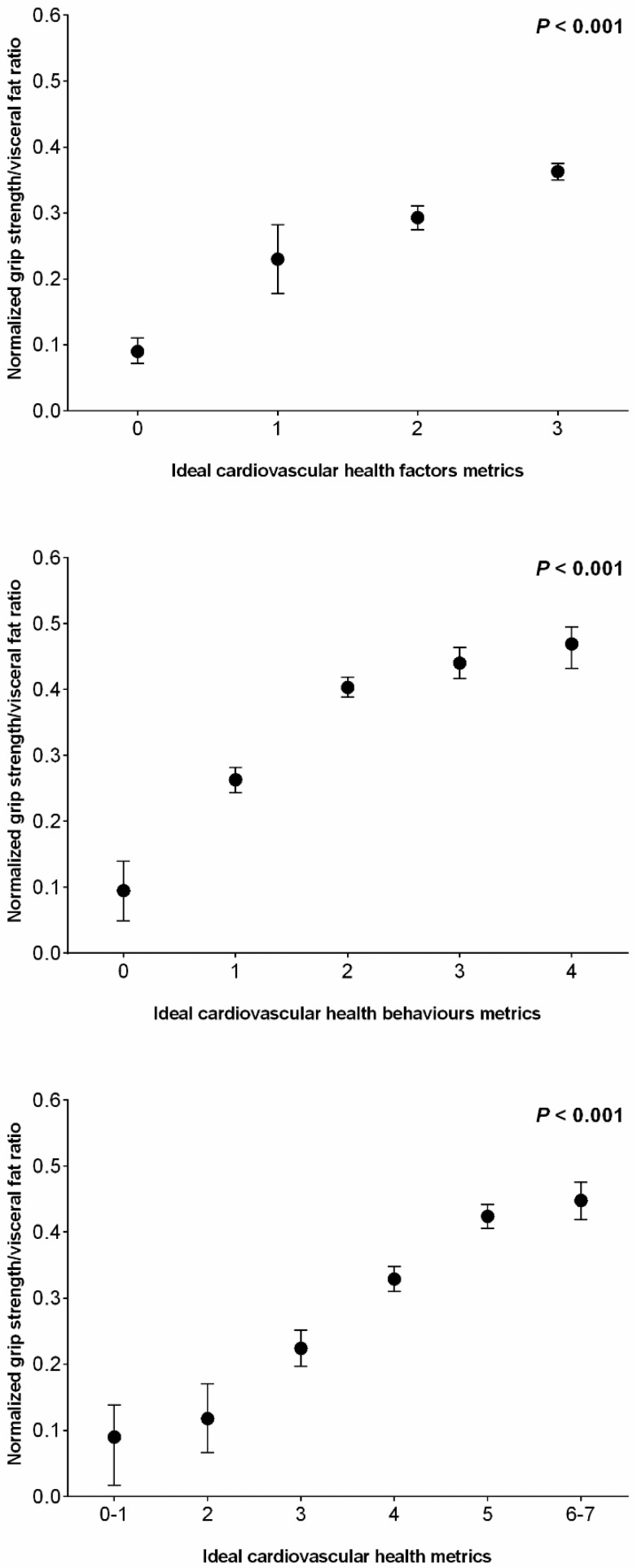
Association between muscular fitness (NGS) to visceral fat ratio across ideal CVH behaviors (smoking, body mass index, cardiorespiratory fitness, and Mediterranean diet adherence), ideal CVH factors (total cholesterol, blood pressure, and plasma glucose) and 7-metrics cardiovascular health in college students.

**Table 1 nutrients-11-00024-t001:** The prevalence rates of the number of cardiovascular health (CVH) metrics, metabolic syndrome (MetS), anthropometric characteristics, blood pressure, and muscular fitness parameters for 1467 study subjects with quartile stratification according to the muscle fitness to visceral fat (MVF) ratio.

**Characteristics**	**1st Quartile** **(*n* = 366)**	**2nd Quartile** **(*n* = 368)**	**3rd Quartile** **(*n* = 366)**	**4th Quartile** **(*n* = 367)**	***p*** **Value** **Chi-Square Test**		
**Goal/Metric**							
Not currently smoking (%)	71.4	74.9	68.9	70.4	0.431		
BMI < 25 kg/m^2^ (%)	11.5	75.5	99.5	100.0	<0.0001		
Physically active (%)	19.1	33.2	24.1	39.2	<0.0001		
Healthy diet (%)	12.4	12.5	10.8	14.5	0.473		
Total cholesterol < 200 mg/dL (%)	93.4	95.6	95.9	96.2	0.088		
Fasting glucose < 100 mg/dL (%)	86.6	91.5	90.7	91.8	0.034		
**Metabolic syndrome** **prevalence**	32.3	5.0	1.9	2.2	<0.0001		
	**1st Quartile** **(*n* = 366)**	**2nd Quartile** **(*n* = 368)**	**3rd Quartile** **(*n* = 366)**	**4th Quartile** **(*n* = 367)**	**ANCOVA ***		
					**Q1 vs. Q2**	**Q1 vs. Q3**	**Q1 vs. Q4**
**Anthropometric**							
Age (years)	21.5 (21.2–21.8)	21.0 (20.8–21.4)	20.0 (19.7–20.3)	19.8 (19.6–20.1)	**-**	**-**	**-**
Weight (kg)	77.2 (76.2–78.1)	64.4 (63.5–65.4)	56.6 (55.6–57.5)	54.7 (53.8–55.6)	<0.0001	<0.0001	<0.0001
Height (m)	1.65 (1.65–1.66)	1.64 (1.64–1.65)	1.63 (1.63–1.64)	1.64 (1.64–1.65)	0.211	0.010	0.178
WC (cm)	85.1 (84.4–85.9)	75.3 (74.6–76.0)	69.4 (68.8–70.2)	67.4 (66.8–68.1)	<0.0001	<0.0001	<0.0001
BMI (kg/m^2^)	28.1 (27.8–28.4)	23.8 (23.5–24.1)	20.9 (20.6–21.2)	20.3 (20.1–20.6)	<0.0001	<0.0001	<0.0001
Fat mass (kg)	23.9 (23.4–24.4)	14.8 (14.4–15.4)	9.6 (9.1–10.1)	8.8 (8.4–9.4)	<0.0001	<0.0001	<0.0001
Fat mass index (kg/m^2^)	8.7 (8.6–9.0)	5.5 (5.4–5.8)	3.5 (3.4–3.8)	3.4 (3.3–3.6)	<0.0001	<0.0001	<0.0001
Skeletal muscle mass (kg)	50.7 (50.3–51.3)	47.0 (46.6–47.6)	44.6 (44.1–45.1)	43.4 (43.0–43.9)	<0.0001	<0.0001	<0.0001
Body fat (%)	30.8 (30.3–31.3)	23.3 (22.9–23.9)	17.4 (16.9–17.9)	16.2 (15.8–16.7)	<0.0001	<0.0001	<0.0001
Visceral fat level	5.6 (5.4–5.8)	2.2 (2.1–2.4)	1.2 (1.1–1.4)	0.8 (0.7–1.0)	<0.0001	<0.0001	<0.0001
MVF ratio	0.08 (0.07–0.09)	0.22 (0.21–0.22)	0.43 (0.42–0.44)	0.60 (0.59–0.61)	<0.0001	<0.0001	<0.0001
**Blood pressure**							
Systolic blood pressure (mmHg)	120.1 (118.8–121.5)	116.6 (115.4–118.0)	112.4 (111.1–113.8)	111.9 (110.7–113.2)	0.001	<0.0001	<0.0001
Diastolic blood pressure(mmHg)	76.1 (74.8–77.3)	74.6 (73.5–75.9)	71.4 (70.3–72.6)	71.2 (70.1–72.3)	0.374	<0.0001	<0.0001
**Muscular fitness**							
Handgrip (kg)	30.9 (30.3–31.6)	30.8 (30.2–31.5)	27.7 (27.0–28.4)	32.3 (31.6–32.9)	1.000	<0.0001	0.033
NGS	0.40 (0.39–0.41)	0.47 (0.46–0.48)	0.48 (0.47–0.49)	0.58 (0.58–0.59)	<0.0001	<0.0001	<0.0001

Values are presented as the mean (95% CI). The MVF ratio was divided into quartiles with the following (min–max) values: Q1: 0.015–0.136, Q2: 0.137–0.338, Q3: 0.339–0.498 and Q4: 0.499–0.904. * To compare between groups, all dependent variables were analyzed by using ANCOVA with adjustment by age, gender, university and alcohol use as covariates. Categorical variables were analyzed by using the Chi-square test. WC, waist circumference; BMI, body mass index; MVF ratio, muscular fitness to visceral fat level ratio; NGS, normalized grip strength (Handgrip (kg)/body mass (kg)).

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
