# Peer review of "Muscle Fitness to Visceral Fat Ratio, Metabolic Syndrome and Ideal Cardiovascular Health Metrics"

_nutrients, 2018, doi:10.3390/nu11010024_

Round 1
Reviewer 1 Report
Thank you for the opportunity to review this manuscript which was very interesting and easy to read. This paper is important as it aims to investigate the association between the muscle fitness to visceral fat (MVF) ratio and the prevalence of metabolic syndrome (MetS) and ideal cardiovascular health (CVH) metrics in a large population 1,464 young adults. I have only a few comments for consideration to strengthen what is already a good manuscript.
-Before the recruitment process, was there a meeting with the participants to explain the aims of the study and the required assessments ?
-Were anthropometric measures performed following the International Standards for Anthropometric Assessment? This information should be included.
-Please specify the WHO cut-off points used in the methods section.
-Add information on the type of BI devices used. For example, are the instruments single- or multiple-frequency or phase-sensitive? How is body composition estimated with these devices? Include reference(s) that support the validity of these different BIA devices to estimate body composition.
Author Response
Reviewer 1
Thank you for the opportunity to review this manuscript which was very interesting and easy to read. This paper is important as it aims to investigate the association between the muscle fitness to visceral fat (MVF) ratio and the prevalence of metabolic syndrome (MetS) and ideal cardiovascular health (CVH) metrics in a large population 1,464 young adults. I have only a few comments for consideration to strengthen what is already a good manuscript.
Author: Thank you for your constructive comments.
-Before the recruitment process, was there a meeting with the participants to explain the aims of the study and the required assessments ?
Author: Thanks for your comment. Before the recruitment process, collegiate students attended an information meeting, where they were informed of the purpose and procedures of the study. This information has been included in the “Methods” section of the revised manuscript (See page 2 lines 85-86).
-Were anthropometric measures performed following the International Standards for Anthropometric Assessment? This information should be included.
Author: We have now included that anthropometric variables were measured in accordance with the International Society for the Advancement of Kinanthropometry (ISAK) guidelines (See page 3 lines 108-109).
-Please specify the WHO cut-off points used in the methods section.
Author: As the reviewer suggested this data has been included in the “Methods” section of the revised manuscript (see page 3 line 103).
-Add information on the type of BI devices used. For example, are the instruments single- or multiple-frequency or phase-sensitive? How is body composition estimated with these devices? Include reference(s) that support the validity of these different BIA devices to estimate body composition.
Author: Thanks for this useful comment. Accordingly, we have added that the single-frequency foot-to-foot BIA device provides BF% by subtracting to the weight of fat-free mass to the total body water. The impedance between the two feet was measured while an alternating current (50 kHz and ∼200 μA) passed through the lower body (See page 3 lines 114-117).
Reviewer 2 Report
Line 81: 1.838 should read as 1,838
Line 91: Please provide reason as to why participants were instructed to wear shorts and a t-shirt for the physical exam.
Body mass index seems to have been described twice, once in section 2.2 and then again in section 2.3 – this can be described only once.
Author Response
Reviewer 2
Line 81: 1.838 should read as 1,838
Author: Thanks for notifying it. We have modified it (See page 2 line 83).
Line 91: Please provide reason as to why participants were instructed to wear shorts and a t-shirt for the physical exam.
Author: In order to provide a precise body weight, participants were requested to wear light clothing (for example t-shirt and shorts). We have specified it in the revised version of the manuscript (See page 3 line 96).
Body mass index seems to have been described twice, once in section 2.2 and then again in section 2.3 – this can be described only once.
Author: Thanks for notifying this mistake. We have removed it in the “Methods” section of the revised manuscript (See page 3 lines 100-103).
Reviewer 3 Report
The work by Ramirez-Velez is well-written and aims to evaluate the association of reduced muscle fitness (normalized grip strength) and increased visceral fat mass ratio and the prevalence of MetS and ideal CVH metrics among college students. In my opinion, this topic of research is interesting, since studies on these relationships, especially in young adults, are scarce. In the context of cardiovascular health, I only suggest to include a brief comment on the deposition of visceral fat in the heart in the introduction or in the discussion sections (doi: 10.3390/jcm7050113) I suggest to add a brief background in the abstract The introduction provides sufficient background and relevant references. Research design is adequate and Methods are well described. My only comment is about the AHA criteria that the Authors used. I suggest to explain if criteria are consistent with those of the AHA, or to motivate changes. I suppose, for example, that criteria used to evaluate diet and physical activity are different from AHA recommendations. Moreover, I suggest to better explain variables included in regression models. Results are clearly presented. My minor comment is the following: Results should be presented and discussed as p-trend. For instance, the sentence “The body weight, waist circumference, and BMI of the subjects in Q1 were significantly higher than those of the subjects in Q2, Q3, and Q4 (P for trend < 0.001)” should be revised as “the was an increasing trend of body weight, waist circumference, and BMI from Q1 to Q4….” (i.e. this in only my proposal). Moreover, I suggest to test p-trend in table 1 (from Q1 to Q4), instead of two-group comparisons (Q1 vs Q2, Q1 vs Q3 and Q1 vs Q4). In general, discussion and conclusion are supported by the results. The Authors well discussed the main limitations of this study, such as the cross-sectional design and the use of BIA. I suggest to include as limitation that the Authors were not able to adjust for unmeasured variables that could affect cardiovascular health, muscle fitness and visceral fat deposition (e.g. doi: 10.1016/j.numecd.2018.04.002).Author Response
Reviewer 3
The work by Ramirez-Velez is well-written and aims to evaluate the association of reduced muscle fitness (normalized grip strength) and increased visceral fat mass ratio and the prevalence of MetS and ideal CVH metrics among college students. In my opinion, this topic of research is interesting, since studies on these relationships, especially in young adults, are scarce. In the context of cardiovascular health, I only suggest to include a brief comment on the deposition of visceral fat in the heart in the introduction or in the discussion sections (doi: 10.3390/jcm7050113). I suggest to add a brief background in the abstract. The introduction provides sufficient background and relevant references. Research design is adequate and Methods are well described. My only comment is about the AHA criteria that the Authors used. I suggest to explain if criteria are consistent with those of the AHA, or to motivate changes. I suppose, for example, that criteria used to evaluate diet and physical activity are different from AHA recommendations. Moreover, I suggest to better explain variables included in regression models. Results are clearly presented. My minor comment is the following: Results should be presented and discussed as p-trend. For instance, the sentence “The body weight, waist circumference, and BMI of the subjects in Q1 were significantly higher than those of the subjects in Q2, Q3, and Q4 (P for trend < 0.001)” should be revised as “the was an increasing trend of body weight, waist circumference, and BMI from Q1 to Q4….” (i.e. this in only my proposal). Moreover, I suggest to test p-trend in table 1 (from Q1 to Q4), instead of two-group comparisons (Q1 vs Q2, Q1 vs Q3 and Q1 vs Q4). In general, discussion and conclusion are supported by the results. The Authors well discussed the main limitations of this study, such as the cross-sectional design and the use of BIA. I suggest to include as limitation that the Authors were not able to adjust for unmeasured variables that could affect cardiovascular health, muscle fitness and visceral fat deposition (e.g. doi: 10.1016/j.numecd.2018.04.002).
Author: Thank you for your constructive comments. As the reviewer indicated, we have included new information regarding the importance epicardial adipose tissue on cardiovascular health in the “Introduction” section of the revised manuscript (See page 3 lines 45-47 and 56-57). In addition, we have explained that although criteria were consistent with those of the AHA, some modifications were required to evaluate diet. This information has been included in the revised version of the manuscript (See page 4 lines 166-168). Also, in order to be more accurate we have explained the variables included in the regression models in the “Statistical analysis” section (See page 5 lines 197-201). As the reviewer indicated, the presentation of results as p-trend has been modified in the revised version (See page 5 lines 211-212). We would like to note that in Table 1 we performed two-group comparisons (Q1 vs Q2, Q1 vs Q3 and Q1 vs Q4) in order to provide a more comprehensive analysis. Finally, according to the reviewer comment we have included as a potential limitation of our study that we were not able to adjust for unmeasured variables such as eating time interval and frequency that could affect cardiovascular health, muscle fitness and visceral fat deposition in the “Discussion” section of the revised manuscript (See page 8 lines 289-291).